# The huntingtin inclusion is a dynamic phase-separated compartment

Fahmida Aktar[1,*], Chakkapong Burudpakdee[2,*], Mercedes Polanco[1,*], Sen Pei[1] , Theresa C Swayne[3], Peter N Lipke[4,5] , Lesley Emtage[1,5]

**Inclusions of disordered protein are a characteristic feature of most neurodegenerative diseases, including Huntington's disease. Huntington's disease is caused by expansion of a polyglutamine tract in the huntingtin protein; mutant huntingtin protein (mHtt) is unstable and accumulates in large intracellular inclusions both in affected individuals and when expressed in eukaryotic cells. Using mHtt-GFP expressed in *Saccharomyces cerevisiae*, we find that mHtt-GFP inclusions are dynamic, mobile, gel-like structures that concentrate mHtt together with the disaggregase Hsp104. Although inclusions may associate with the vacuolar membrane, the association is reversible and we find that inclusions of mHtt in *S. cerevisiae* are not taken up by the vacuole or other organelles. Instead, a pulse-chase study using photo-converted mHtt-mEos2 revealed that mHtt is directly and continuously removed from the inclusion body. In addition to mobile inclusions, we also imaged and tracked the movements of small particles of mHtt-GFP and determine that they move randomly. These observations suggest that inclusions may grow through the collision and coalescence of small aggregative particles.**

## Introduction

Large aggregates of unfolded protein are a common characteristic of neurodegenerative diseases. Protein aggregation begins with the association of small numbers of unfolded polypeptides, and is believed to proceed through oligomeric intermediate species to form large, visible aggregates in vivo (Thakur et al, 2009; Tyedmers et al, 2010; Arosio et al, 2014). Aggregates can assume different forms, from relatively unstructured, such as in diffuse Aβ plaques, to highly structured, such as found in amyloid aggregates.

Huntington's disease is caused by expansion of glutamine repeats near the N terminus of the huntingtin protein (Htt). Normal human Htt contains between 9 and 37 glutamines in the repeat

tract, whereas pathogenic forms contain ≥37 glutamines in the repeat region (Kieburtz et al, 1994). Mutant huntingtin (mHtt) undergoes cleavage to release an N-terminal fragment containing the polyglutamine region (Becher et al, 1998; Thomas et al, 1998; Landles et al, 2010), and this fragment forms inclusions in individuals affected by Huntington's disease (Scherzinger et al, 1997; Huang et al, 1998; McGowan et al, 2000).

We are interested in studying the mechanism by which inclusions are formed and removed in budding yeast model system, *Saccharomyces cerevisiae*. Exon 1 of huntingtin, with an expanded polyglutamine region and fused to GFP, has been used to model inclusion formation in eukaryotic cells, including yeast (Mason & Giorgini, 2011). This fragment intrinsically forms visible aggregates upon gene expression, even without heat shock or other stress. *S. cerevisiae* cells expressing mHtt-GFP have been shown biochemically to contain insoluble mHtt and cytoplasmic inclusions are visible by fluorescence microscopy (Krobitsch & Lindquist, 2000; Muchowski et al, 2000; Dehay & Bertolotti, 2006; Mason & Giorgini, 2011). mHtt-GFP inclusions in yeast have been variously described as localized to an aggresome through microtubule-based transport (Wang et al, 2009), or in perivacuolar insoluble protein deposits (IPODs), which contain autophagosomal protein Atg8, do not exchange material with the cytoplasm, and are suggested to be removed into the vacuole by autophagy (Kaganovich et al, 2008).

Here, we use high-resolution optical techniques to investigate the nature of the mHtt-GFP inclusion and the fate of material within it. We find that the mutant Htt inclusion has properties different from IPODs or aggresomes. The inclusion is a mobile compartment, and its contents can move about freely within it. Moreover, its contents are in flux, with mHtt being measurably incorporated and released on a timescale of minutes to hours. In addition, we observe small cytosolic mHtt aggregates coexisting with the large inclusion. These small aggregates move randomly, without any contribution from active transport. Taken together, our results suggest a model in which the mobile, gel-like mHtt inclusion body (IB) in *S. cerevisiae* grows as a result of collision and coalescence with diffusing aggregate particles.

---

[1]Biology Department, City University of New York, York College, Queens, NY, USA    [2]Department of Biology, Pace University, New York, NY, USA    [3]Herbert Irving Comprehensive Cancer Center, Columbia University, New York, NY, USA    [4]Biology Department, City University of New York, Brooklyn College, Brooklyn, NY, USA    [5]Molecular, Cellular and Developmental Biology Program, City University of New York Graduate Center, New York, NY, USA

Correspondence: lemtage@york.cuny.edu
*Fahmida Aktar, Chakkapong Burudpakdee, and Mercedes Polanco contributed equally to this work

 

# Results

## Mutant Htt forms several types of aggregates in yeast

To understand the mechanisms of protein aggregate formation and clearance, we first sought to more completely characterize inclusions formed by mHtt(72Q)-GFP in yeast. Previous work has shown that cells expressing native Htt(25Q)-GFP do not form inclusions at all, and that the propensity for inclusion formation in yeast increases with the length of the polyglutamine repeat sequence, as it does in other model systems (Krobitsch & Lindquist, 2000; Muchowski et al, 2000). In humans, Htt is widely expressed throughout life; mutant Htt constitutes an ongoing burden on cells, unlike unfolded protein resulting from transient stresses. Therefore, we expressed exon 1 of

mHtt(72Q)-GFP or mHtt(25Q)-GFP (Krobitsch & Lindquist, 2000) constitutively from a low-copy CEN plasmid, typically present at 1–5 copies per cell (Karim et al, 2013), under the glycerol phosphate dehydrogenase promoter. As inclusion frequency and size vary significantly with growth phase (Fig S1), all of our observations are carried out on overnight cultures in mid-log phase.

Consistent with earlier studies, we do not observe aggregates in cells expressing native Htt(25Q)-GFP (n = 84). In contrast, in cells expressing mHtt(72Q)-GFP, quantitative epifluorescence and confocal microscopy reveal diversity in Htt inclusion number and structure. We determined the size, prevalence, and localization of inclusions.

In cells that contain an inclusion, mHtt(72Q)-GFP is typically found in a single IB. Most IBs are ovoid, with smooth edges and uniform GFP intensity throughout the structure (Fig 1A). These ovoid

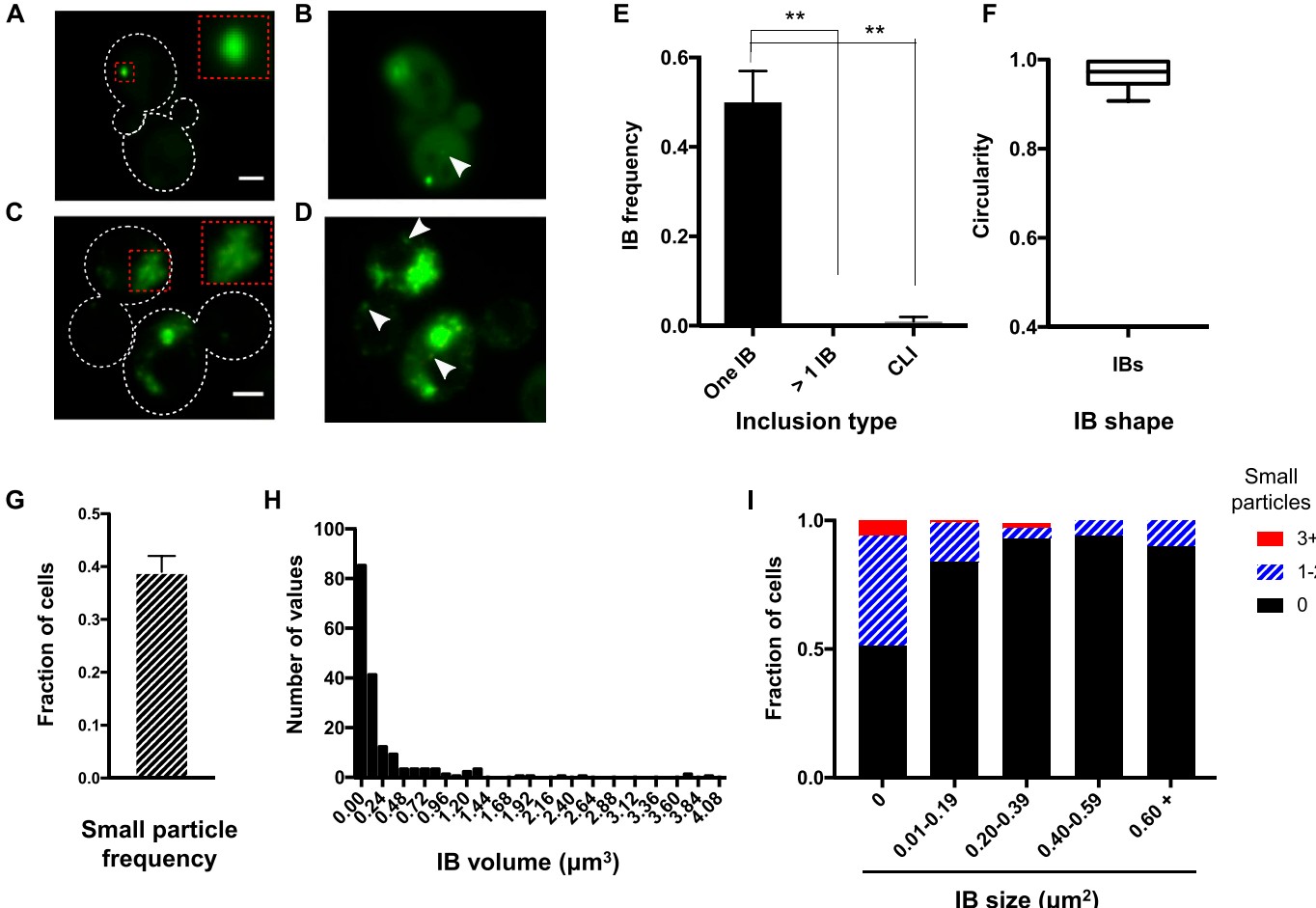

**Figure 1.  Mutant Htt typically forms a single ovoid IB, accompanied by small particles.**
**(A)** Mutant Htt(72Q)-GFP in mid-log–phase cells forms a single, ovoid, uniformly fluorescent IB. A single optical section is shown. Cell outlines are indicated by dotted white lines. Inset shows enlarged version of area indicated by dotted red rectangle. Bar, 2 $\mu$m. **(B)** Small particles are sometimes seen (arrowhead); a single optical section is shown, from the same z-series as in (A), but in a different imaging plane and with enhanced contrast. **(C)** In a small fraction of cells, mHtt(72Q)-GFP is found in asymmetric CLIs. Cell outlines are indicated by dotted white lines. Inset shows enlarged version of area indicated by dotted red rectangle, showing inhomogeneous intensity of the CLIs. A single, representative optical section is shown. Bar, 2 $\mu$m. **(D)** Numerous smaller inclusions often accompany a CLI (arrowheads). A single optical section is shown, from the same z-series as in (C), but in a different imaging plane and with contrast enhanced to reveal the smaller inclusions. **(E)** In mid-log cells, single IBs are significantly more prevalent than either multiple IBs or CLIs. Cells were in mid-log phase and had been growing for at least 10 doubling times. The mean of three trials is shown, n = 166 cells total. Error bars represent SEM. **P < 0.01, ANOVA and unpaired two-tailed t test. **(F)** Ovoid IBs are approximately circular (whiskers indicate minimum and maximum values; n = 16 IBs with diameters over 0.4 $\mu$m). **(G)** Quantitation of the frequency of small particles in the same cells quantified in (E); error bars indicate SEM. **(H)** Histogram of apparent IB volumes calculated from 3D series (n = 184). **(I)** The fraction of cells containing 0, 1–2, or 3 or more small particles was determined in cells with different IB sizes (defined as maximal apparent cross-sectional area). n = 48–233 cells per category.

IBs in yeast are similar in appearance to the mHtt inclusions formed in mammalian cells (Arrasate et al, 2004; Yamamoto et al, 2006). The volume of IBs varies widely, although most IBs have an apparent volume <0.2 $\mu m^3$ (Fig 1H). In rare cases, cells contain more than one inclusion. Many cells without IBs contain one or more visible small cytoplasmic particles; some cells contain both small particles and ovoid IBs (Fig 1B and I).

A minority of cells contain different forms of inclusion. The most common alternate form is a cluster of smaller inclusions, which are asymmetric and variable in intensity throughout (Fig 1C). Cluster-like inclusions (CLIs) are invariably accompanied by large numbers of smaller, intense inclusions, or peripheral aggregates, distributed widely throughout the cytoplasm (Fig 1D) and are found in 1–2% of cells (Fig 1E). Interestingly, the presence of CLIs correlates with a reduced intensity of mHtt-GFP in the cytoplasm, compared with that in cells containing an ovoid IB. The average cytoplasmic GFP intensity is 148 ± 12 A.U. for cells with IBs, compared with 29 ± 4 A.U. for cells with CLIs (SEM, n = 61, 6 respectively).

Whereas CLIs are asymmetrical in intensity and shape, the predominant species of inclusion, ovoid IBs, appear to be spherical, or nearly so. A commonly used measure of circularity is the circularity ratio (CR), which is calculated as $4\pi \times area/(perimeter)^2$; for a perfect circle, CR = 1. We find that ovoid IBs have an average CR of 0.97 ± 0.007 (Fig 1F), and an aspect ratio of 1.15 ± 0.02 (SEM, n = 18).

About half of cells in cultures that are in mid-log phase and have been continuously growing for 16 or more hours have an IB (Figs 1E and S1). Since about half the cells are new daughter cells and daughter cells do not usually inherit Htt inclusion bodies (manuscript in preparation), the cells containing IBs are predominantly one or more generations old. Therefore, most IBs form within about one cell cycle, which is ~100 min under the growth conditions used in this study. Indeed, time-lapse imaging of cells growing on agar pads made with growth medium shows that the average time for a newly budded cell to form an IB is 90 ± 10 min (mean ± SEM, n = 16), with an average cell division time of 120 ± 4 min (n = 38).

In addition to IBs, about 40% of cells expressing Htt(72Q)-GFP also contain small aggregates (Fig 1G). There is an inverse relationship between the presence and size of an IB and the number of small particles detected in the cell (Fig 1I). One possible explanation for the apparent decline in number of small particles with IB size is that small particles are removed from the cytoplasm through integration into the IB.

### Inclusion bodies and small particles of mHtt are mobile

When imaging living cells expressing mHtt-GFP, we observe movement of both IBs and small particles of mHtt. Time-lapse imaging demonstrates that both structures are highly dynamic: moving, and frequently changing direction. Very large IBs are more constrained in their movement (Fig 2A and B and Videos 1 and 2). In contrast, CLIs are immobile, perhaps because of steric hindrance due to their large size.

Cells typically contain single large IBs and a few small particles. The inverse relationship between the presence and size of an IB and the number of small particles (Fig 1I) suggests that small particles may be integrated into the growing IB. Conceivably, this integration could occur through directed movement, that is, on

cytoskeletal tracks, or alternatively it could occur through random collision. We, therefore, sought to ascertain whether the movement of small aggregates through the cytosol is random or directional. The movement of small particles is too rapid to be resolved in 3D by standard confocal microscopy. However, using spinning-disk confocal imaging, we have been able to analyze the movement of small aggregates in 2D, as they randomly enter and move within the focal plane (Fig 2B and C).

To determine the degree of directedness of small particles, we determined their displacement over different time intervals. For a 2D random walk, the mean squared displacement $(\Delta d)^2$ of a particle exhibiting Brownian motion varies directly with time. In general, we can say that

$$(\Delta d)^2 \propto (\Delta t)^\alpha$$

where a value of $\alpha = 1$ indicates random diffusion, and a value $\alpha > 1$ indicates a degree of superdiffusion, indicative of an active transport process.

Analysis of the motion of small particles reveals that for small particles, $\alpha = 1.00 \pm 0.04$ (SEM, n = 23), consistent with particles that are freely diffusing (Fig 2D and F). The apparent diffusion constant is $5.9 \times 10^{-2} \pm 0.7 \times 10^{-2}\ \mu m^2/s$, a value slightly larger than the diffusion constant of a ribosome in *Escherichia coli* (Bakshi et al, 2012), suggesting either a mass on the order of a ribosome, or an eccentric shape, such as that of an amyloid fibril, or both. In contrast, the average value of $\alpha$ for IBs is 1.2 ± 0.03 (SEM, n = 21), indicating that IBs are being actively transported (Fig 2E and F).

### Unfolded protein in the inclusion is mobile

Our analysis of small mHtt aggregate dynamics raises the possibility that small particles of mHtt, diffusing through the cytosol, are added to the IB through random collision. This model is supported by the observation that the small particles are detected less often as the surface area of the IB increases (Fig 1I). In an effort to understand how the IB grows, we used FRAP to visualize the addition of new material to IBs.

If the IB grows through accretion of monomers or small aggregates to the outer surface of the IB, we would predict fluorescence recovery to occur at the periphery of a photobleached IB. Furthermore, a previous study reported that mHtt inclusions do not recover fluorescence within 20 s of photobleaching (Peskett et al, 2018), suggesting that the IB is not liquid. However, diffusion of tagged proteins from gels in vitro has been shown to occur on much longer timescales, minutes to hours (Kato et al, 2012). Therefore, we followed recovery longer and imaged less frequently to reduce unwanted photobleaching during recovery. Surprisingly, we find that fluorescence begins to recover within 30 s, and the recovered fluorescence is present throughout the entire structure (n = 18; Fig 3A). This suggests that the contents of the IB are able to diffuse within it, and that the interior of the IB has liquid-like or gel-like properties.

To assess our ability to resolve the possible accretion of a layer of protein around an inclusion, we used cells expressing both mHtt-GFP and Vph1-mCherry to simultaneously visualize the IB and the vacuolar membrane (Fig 3B–E). We are able to clearly resolve the

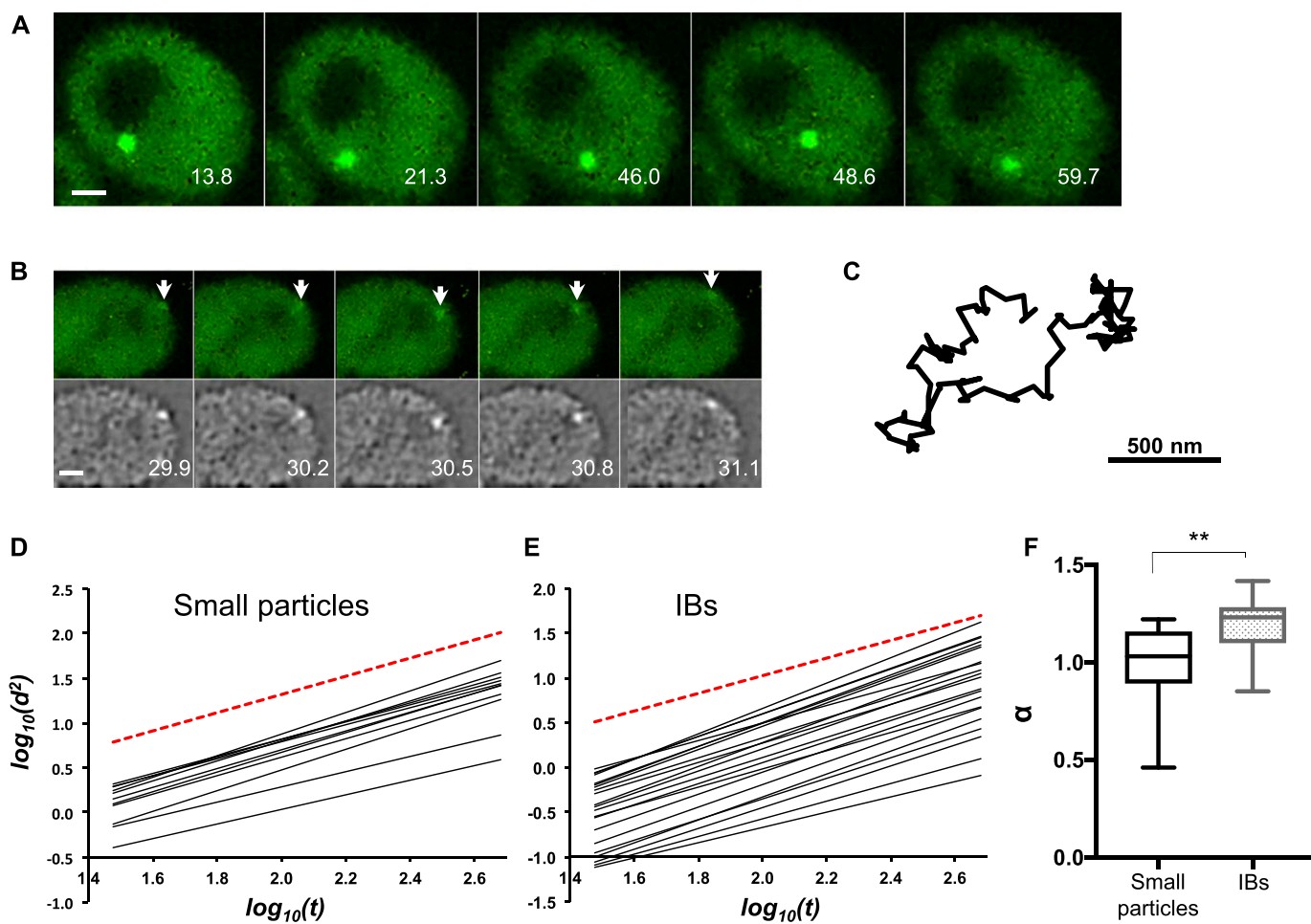

**Figure 2. Inclusions and small particles of aggregated mHtt are mobile.**
**(A)** Representative time-lapse sequence of an IB moving within the cytoplasm. Numbers indicate elapsed time in seconds. Bar, 1 $\mu$m. **(B)** Representative time-lapse frames showing a small particle (arrows). Upper panels, denoised images; lower panels, the same images processed with the SpotTracker Spot Enhancing Filter 2D. Numbers indicate elapsed time in seconds. Bar, 1 $\mu$m. **(C)** Track of the particle indicated in (B). **(D, E)** The log of mean squared displacement ($d^2$) is plotted against the log of time (t) for 23 small particles (D) or 23 IBs (E). Each fitted line is derived from a single small particle. The slope of the line gives the value for the exponent $\alpha$ in the diffusion equation. The red dashed reference line shows $\alpha$ = 1 (random diffusion). A slope >1 indicates directed movement. **(F)** The average value of $\alpha$ was determined for small particles and IBs. **P = 0.0006, unpaired two-tailed t test with unequal variance, n = 23 for each group; whiskers indicate minimum and maximum values.

sides of a vacuole with a diameter of ~400 nm in cells expressing Vph1-mCherry (Fig 3D and E). Fluorescence recovery of an mHtt(72Q)-GFP inclusion with a diameter of about 1 $\mu$m, present in the same cell as the vacuole shown in Fig 3B, is shown in Fig 3D and E. The newly added mHtt(72Q)-GFP accumulates inside the inclusion rather than adding to the perimeter. The material within the post-bleach IB appears to be distributed inhomogeneously, but the brighter regions are not concentrated at the periphery of the IB (Fig S2A and B). Even though the extensive photobleaching needed to extinguish the IB also bleaches the cytoplasm, the apparent IB diameter recovers to 50% of the pre-bleach value within ~4 min (Fig 3F). Thus, the results demonstrate the highly dynamic and mobile nature of Htt within the cytoplasm and in the IB.

We considered the possibility that fluorescence recovery was due to some other change in the fluorophore, such as dequenching, within the IB itself and did not represent the addition of new material. If that were the case, the intensity of the recovered fluorescence should be independent of the level of post-bleach cytoplasmic fluorescence. However, we find that the total amount of recovered fluorescence correlates strongly with the post-bleach cytoplasmic intensity and does not correlate with pre-bleach IB total fluorescence (Fig S3A and B), supporting the conclusion that the fluorescence recovery in the IB is due to addition of mHtt(72Q)-GFP from the cytoplasm.

### Htt protein is exchanged between the inclusion and the cytoplasm

We next sought to determine the fate of mHtt-GFP after entering the IB. Despite the rapid flow of cytoplasmic protein into IBs, time-lapse imaging shows that IBs are not removed from cells at an observable rate. We performed 3D imaging of 120 individual mHtt IBs of different sizes for an average of 3 h each, at 10-min intervals. In this total of >21,000 min of tracking, no IB ever disappeared from a cell. Therefore, if mHtt inclusion bodies are removed from actively growing, unstressed cells, it must be a rare event.

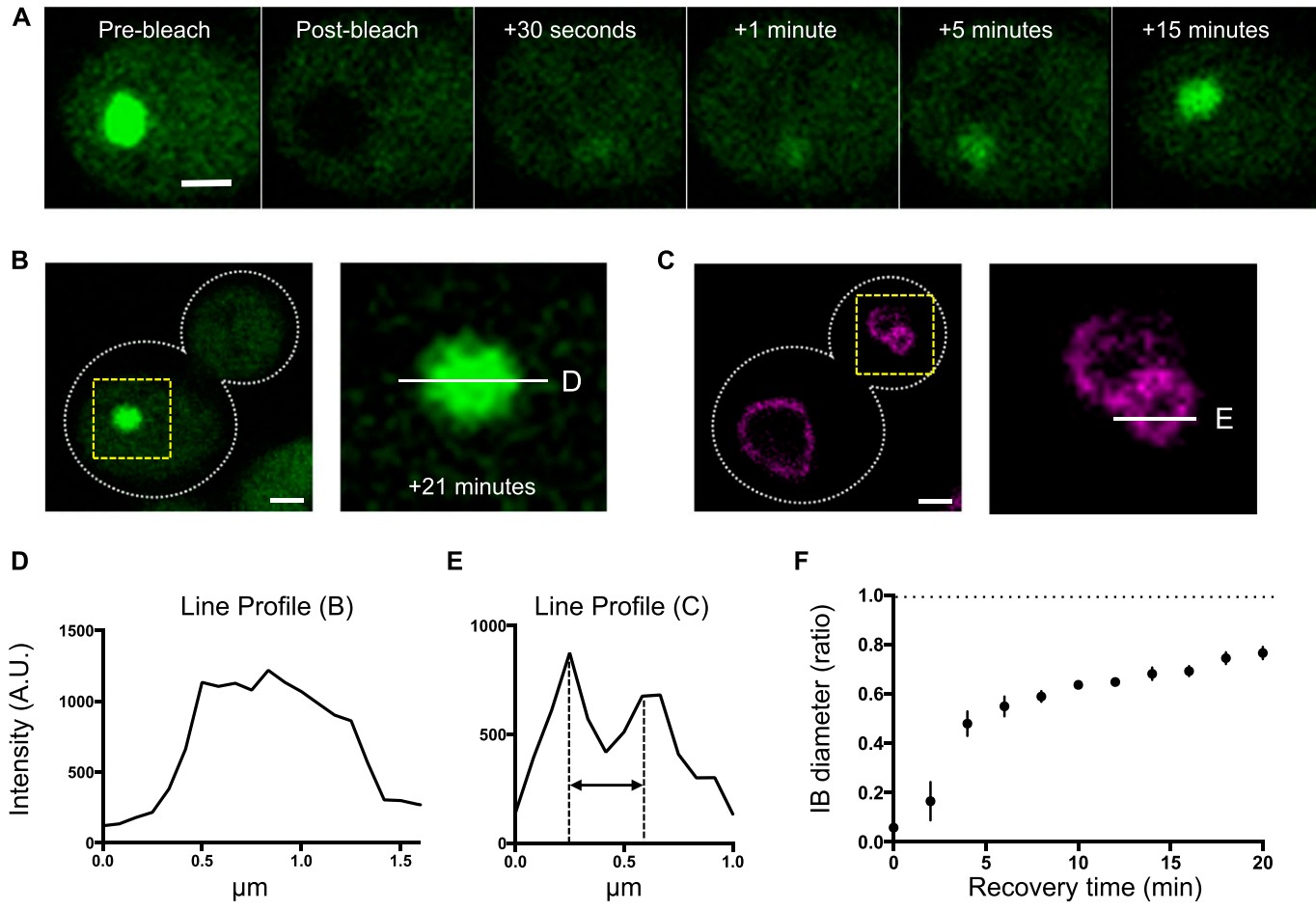

**Figure 3. The mHtt within an IB is mobile.**
**(A)** Recovery of mHtt(72Q)-GFP fluorescence within an IB following photobleaching. Single planes are shown from z-series taken at 200-nm intervals. Bar, 1 $\mu$m. In the pre-bleach image, the long axis of the IB is 1,100 nm, and the short axis is 900 nm; after 15 min of recovery, the long and short axes of the IB are 900 and 730 nm, respectively. Bar, 1 $\mu$m. **(B, D)** Left panel, single plane showing the same cell as in (A), at 21 min post-bleach. Right panel, enlarged view of area inside yellow box, with white line indicating the location of the profile plot in (D). Bar, 1 $\mu$m. **(C)** Left panel, single plane showing Vph1-mCherry in the same cell as in (A) and (B). Bar, 1 $\mu$m. **(E)** Right panel, enlarged view of area inside yellow box, with white line indicating the location of the profile plot in (E). **(D)** Intensity profile of the line across the center of the IB shown in (B). **(E)** The intensity profile of the line across the center of a small vacuole in the bud shown in (C). Double-headed arrow indicates the distance between the two sides of the vacuole (380 nm). **(F)** Ratio of IB diameter to pre-bleach IB diameter as a function of time of recovery (n = 6, error bars indicate SEM).

To ascertain whether mHtt(72Q) that enters an IB is permanently or transiently incorporated into the IB, we tagged mHtt(72Q) with the photoconvertible protein mEos2. Newly synthesized mEos2 fluoresces green; illumination of the green form with light at 405 nm converts it irreversibly to a red-fluorescent form. The movement of mHtt-mEos2 aggregates after photoconversion is like that of mHtt-GFP aggregates: the average value of $\alpha$ for small particles is 1.04 ± 0.06 (SEM, n = 8) and for IBs is 1.37 ± 0.07 (SEM, n = 8). We photoconverted mEos2 in cytoplasm and IBs in cells containing moderate to large (0.3–1 $\mu$m radius) mHtt(72Q)-mEos2 inclusions, and followed the cells by time-lapse 3D imaging every 10 min for over 4.5 h (Fig 4A).

We find that photoconverted red mHtt(72Q)-mEos2 is depleted from the IB over the course of the experiment. As expected, no new red mHtt(72Q)-mEos2 is produced during the time course. If protein in the IB remains there stably, we would predict that the red fluorescence in the IB would increase as photoconverted protein enters the IB from the cytosol. The red signal would continue to rise until cytosolic levels of red protein are depleted. The total red fluorescence in the IB should then plateau because no additional red protein would be made by the cell or removed from the IB.

Because cytoplasmic red mEos2 remains detectable for the entire time course, we predicted that the amount of red protein in the IB would continue to rise throughout the experiment. Indeed, time-lapse microscopy shows an initial increase in red fluorescence in the IB, but even while red protein levels in the cytoplasm remain at 75–80% of initial levels, the amount of red fluorescence in the IB plateaus. The red fluorescence in the IB then decreased at a rate greater than that caused by photobleaching (Fig 4B). After correction for photobleaching, the total amount of red protein in IBs decreases significantly over 4.5 h of imaging, with a 35% drop between the maximum at 60 min and the final time point 2.5 h later (n = 17, $P$ = 0.0001, two-tailed $t$ test with unequal variance). These findings demonstrate that the mHtt(72Q)-mEos2 protein that is incorporated into the IB is either degraded in the IB itself, or released from the IB.

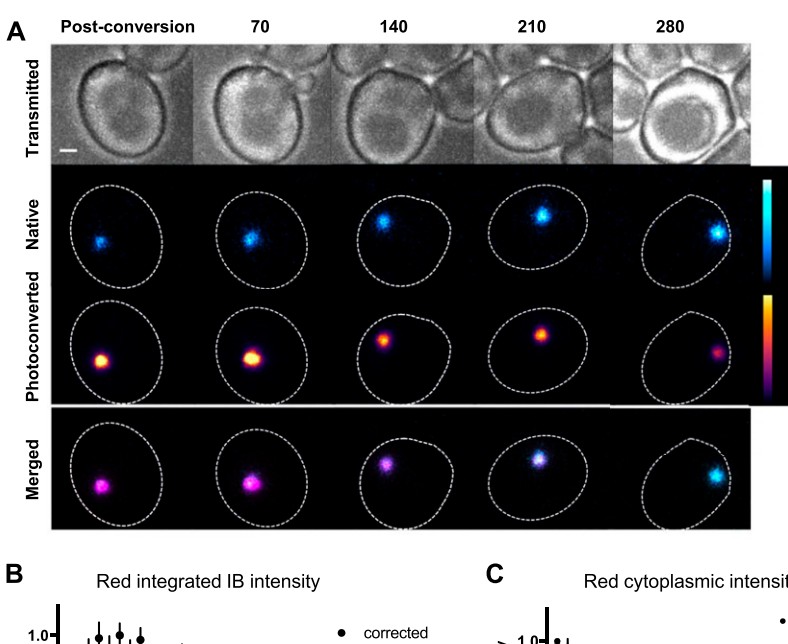

**A** Post-conversion    70    140    210    280

Transmitted

Native

Photoconverted

Merged

**Figure 4.  Mutant Htt in the IB is turned over.**
**(A)** Time-lapse images of mHtt(72Q)-mEos2 were collected every 10 min following photoconversion of a portion of the mEos2 from green to red. Images of a cell containing a large IB taken immediately after photoconversion, and then every 70 min for 280 min, are shown in bright-field (upper panels), the native and photoconverted channels separately (middle two rows), and with native (cyan) and photoconverted (magenta) channels merged (lower panels). Numbers indicate minutes after photoconversion. To better display intensity changes, "fire" and "ice" lookup tables are used in the middle panels; intensity calibration bars are shown on the right. Bar, 1 μm. Mother cell body outlined in dashed white line. **(B, C)** Integrated red intensity (I.I.) of IBs (B) and mean red cytoplasmic intensity (C) after photoconversion for mHtt(72Q)-mEos2–expressing cells was measured over 4.5 h. Values before (grey circles) and after (black circles) correction for photobleaching were normalized and reported as the fraction of the maximum value following photoconversion. The predicted decrease in cytoplasmic intensity based on dilution of the cytoplasm due to cell growth and division is shown (triangles). Mean ± SEM is shown for 17 cells.

**B**  Red integrated IB intensity

• corrected
• uncorrected

**C**  Red cytoplasmic intensity

• corrected
• uncorrected
▲ dilution alone

The average number of cell cycles during each experiment is 2.2, and the average relative size of the daughter cell at division is 70% of the mother cell. Consequently, the photoconverted mEos2 is diluted by a factor of 3.7, which we predict would reduce the concentration, and intensity, to 27% of the original intensity. Instead, after correcting for photobleaching, we see a reduction to 13% of the original intensity, suggesting that red mHtt(72Q)-mEos2 protein is being degraded (Fig 4C).

Conversely, the intensity of green mEos in the IB and cytoplasm rises throughout the time course, as production is greater than the loss due to photobleaching (Fig S4). The green signal cannot accurately be corrected for photobleaching because newly synthesized protein is also green. Therefore, at any time point, the cell contains a heterogeneous population of green mEos2: molecules of diverse ages, which have been exposed to variable numbers of excitations. However, even without correction for photobleaching, the increased green protein levels in the IB are evident. Therefore, the results of the pulse-chase experiment support the idea of dynamic incorporation of mHtt(72Q)-mEos2 into the IB, and concomitant dissociation of the same protein from the IB.

### Hsp104 is required for formation of both small particles and inclusion bodies

Hsp104 is a member of the AAA+ disaggregase family of chaperones and has been shown to unfold aggregated and amyloid protein (Glover & Lindquist, 1998; Goloubinoff et al, 1999; Zolkiewski, 1999;

Zolkiewski et al, 2012; Doyle et al, 2013; Mokry et al, 2015; Gates et al, 2017). Surprisingly, given its disaggregase activity, Hsp104 is required for the formation of mHtt IBs in *S. cerevisiae* (Krobitsch & Lindquist, 2000; Meriin et al, 2002). We confirmed this dependence by introducing mHtt(72Q)-GFP into an *hsp104Δ* strain. As expected, mHtt(72Q)-GFP did not form IBs in this strain. Moreover, we did not observe any small particles of mHtt(72Q)-GFP in the *hsp104Δ* strain (n = 101 mid-log–phase cells), demonstrating that Hsp104 is required for the formation of both small mHtt particles and IBs in yeast (Fig 5A).

Hsp104 is present in unfolded protein inclusions (Kawai et al, 1999; Kaganovich et al, 2008; Saarikangas and Barral, 2015). Therefore, we tagged *HSP104* at its genomic locus with mCherry to ascertain whether Hsp104 is present in small particles as well. Cells expressing native Htt(25Q)-GFP do not form IBs containing Htt. Nonetheless, Hsp104-mCherry is found in intense ovoid IB-like structures in 10% of these cells, perhaps attesting to its role in forming inclusions with other unfolded proteins (Fig 5B and C). In cells expressing mutant Htt(72Q)-GFP, 49% of cells expressing the Hsp104-mCherry fusion contained mHtt IBs, similar to the wild-type parent BY4741 (compare Fig 1E to Fig 5C), demonstrating that the Hsp104 fusion protein does not alter mHtt inclusion formation. Consistent with previous results, all mHtt IBs contained Hsp104p, and IBs containing Hsp104-mCherry were more frequent in cells expressing mutant huntingtin (mHtt(72Q)-GFP) compared with native Htt(25Q)-GFP (54% versus 10%, Fig 5C). Hsp104-mCherry is sometimes seen in IBs that do not appear to contain mHtt-GFP (Fig 5).

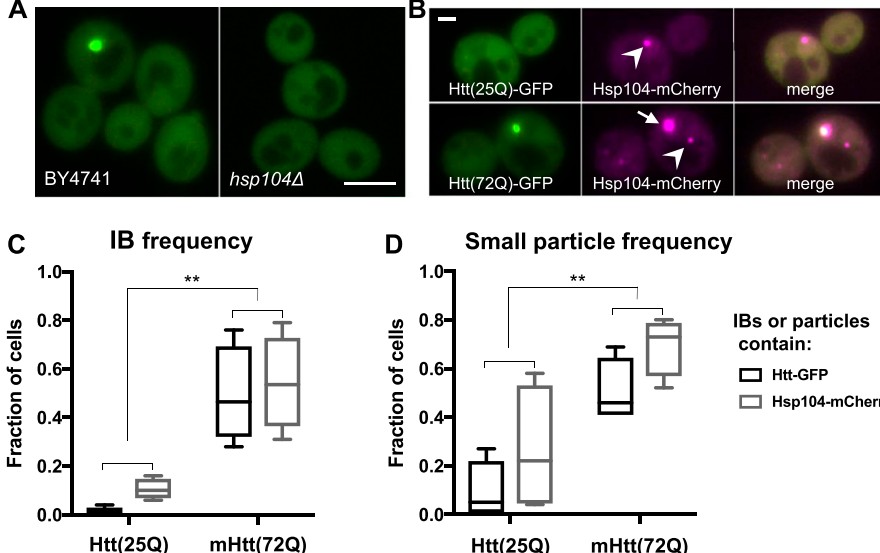

**Figure 5. Hsp104 is found in mHtt particles and is required for their formation.**
**(A)** Representative images of mHtt(72Q)-GFP in wild-type (left panel) and *hsp104Δ* (right panel) cells. Bar, 4 μm. **(B)** Cells containing genomically tagged Hsp104-mCherry and either native Htt(25Q)-GFP (upper panels) or mHtt(72Q)-GFP (lower panels). Hsp104-mCherry is displayed as magenta. Arrow, mHtt IB containing Hsp104; arrowheads, Hsp104 IB with no detectable mHtt. Bar, 1 μm. **(C, D)** Cells from the strains shown in (B) were assessed for the presence of IBs and small particles. The fraction of cells containing one or more IBs (C) or small particles (D) bearing the indicated marker was determined. **P = 0.0001–0.0003 for differences between Htt alleles, two-way ANOVA, n = 4 replicates, 124–130 cells total per group; whiskers indicate minimum and maximum values.

This observation is consistent with the occasional formation of Hsp104 IBs in cells expressing native Htt, but also indicates that IBs containing different sets of proteins can form independently in the same cell.

Furthermore, we detect small cytosolic particles containing Hsp104-mCherry in about 26% of cells expressing native Htt(25Q)-GFP and 69% of cells expressing mHtt(72Q)-GFP (Fig 5C and D). In both strains, small particles of Hsp104 are considerably more prevalent than small particles of Htt, suggesting that Hsp104 forms small particles by itself or with other unfolded proteins as well as with mHtt.

The increased prevalence of small Hsp104 particles in cells expressing Htt(75Q)-GFP is consistent with the model that the small particles of mHtt contain both proteins. To definitively address that question, we used spinning-disk confocal microscopy to image living cells co-expressing Hsp104-mCherry and mHtt(72Q)-GFP, and determined that both proteins are present together in small particles (Video 3A–C). Not every mHtt-GFP aggregate also shows detectable Hsp104-mCherry. However, this may be due to the relatively low signal from this fusion protein combined with the short exposure times necessitated by the fast motion of the particles.

### Rnq1 is also required for formation of small particles and inclusion bodies of mHtt(72Q)-GFP

Recent work by Peskett and colleagues describes a liquid-like, phase-separated mHtt assembly in yeast with a different genetic background from that used our studies (Peskett et al, 2018). Although we also provide evidence for phase separation in a biomolecular condensate containing mHtt, the intensely bright, phase-separated mHtt inclusions that we observe in the standard laboratory strain BY4741 are different from those described by Peskett et al (2018). The dim, liquid-like assemblies described by Peskett and colleagues were observed in a strain that had been selected for abnormal Rnq1-GFP behavior. Rnq1 is a prion-like low-complexity protein (LCR) that has been shown to be required for the formation of mHtt-GFP inclusions (Meriin et al, 2002). Peskett and colleagues selected cells in which Rnq1-GFP could no longer aggregate when overexpressed, as it does in most strains (Sondheimer & Lindquist, 2000; Manogaran et al, 2010; Peskett et al, 2018). In contrast, our studies were conducted in an unselected BY4741 strain. Our results are consistent with the initial characterization of *rnq1* deletion strains: cells lacking Rnq1 protein no longer form mHtt-GFP inclusions (Meriin et al, 2002).

As *hsp104Δ* null cells form neither inclusions nor small particles of mHtt, we were interested in ascertaining whether cells without Rnq1, which rarely form IBs, contain small particles of mHtt-GFP. We observe that *rnq1Δ* strains expressing mHtt(72Q)-GFP, similar to *hsp104Δ* strains, typically lack both large inclusion bodies and small particles; no other forms of mHtt(72Q)-GFP aggregate nor liquid assembly were seen in a survey of 195 of these cells (Fig S5).

### mHtt is found in a membrane-less compartment

Our photobleaching data showed that mHtt-GFP inside the mHtt IB is able to diffuse freely, raising the possibility that the IB is a membrane-bound compartment. We used transmitted electron microscopy, combined with immunogold labeling, to examine the ultrastructure of the mHtt IB. The parental strain BY4741 and cells expressing mHtt(72Q)-GFP were fixed and sectioned. Sections were labeled with anti-GFP antibody, followed by a gold particle–conjugated secondary antibody, and then stained with uranyl acetate and lead citrate.

Approximately 5% of sections taken from cells expressing mHtt(72Q)-GFP displayed clusters of gold particles in the cytosol (Fig 6). The size of the clusters, and the fraction of cells with clusters of antibody-labeled structures, were consistent with the expected size and frequency of sectioning through an IB. Control sections from cells that did not express GFP contained occasional, isolated gold particles, but never clusters.

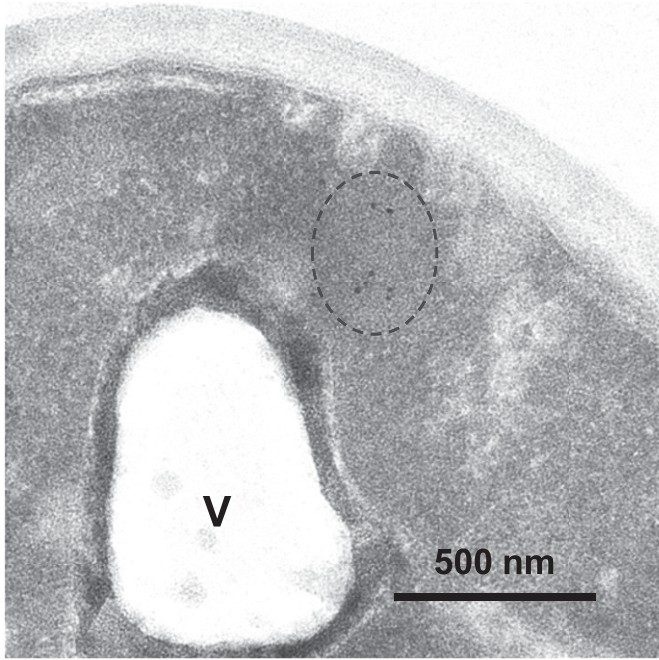

**Figure 6.  Mutant Htt IBs are non–membrane bound.**
Transmitted electron micrograph of a cell expressing mutant Htt(72Q)-GFP, fixed, and stained with anti-GFP and 10-nm gold particle–conjugated secondary antibody. Dotted line indicates a cluster of gold particles, ~500 nm in diameter, in the cytoplasm. V, a lobe of the vacuole, surrounded by membrane. Bar, 500 nm.

Membranes surrounding the vacuole, nucleus, mitochondria, and vesicles were visible (Fig S6), but gold clusters were never surrounded by a membrane, suggesting that this compartment is not membrane bound. Other non–membrane-bound liquid cytosolic compartments, including RNA particles and stress granules, have been described. The smooth edges and ovoid shape of the mHtt IB are reminiscent of these other non–membrane-bound compartments, including P granules, processing bodies, stress granules, as well as more gel-like condensates such as made by the phenylalanine-glycine repeats found in nucleoporins (Brangwynne et al, 2009; Kroschwald et al, 2015; Gopal et al, 2017; Woodruff et al, 2018).

### IBs are not stably associated with microtubule organizing centers (MTOCs) or autophagosomes

The localization of an inclusion is important to its classification (Sontag et al, 2014). In 2008, Kaganovich and colleagues suggested that mutant Htt is incorporated into IPODs, perivacuolar structures containing Atg8, but that it does not localize to the MTOC, often referred to as the spindle-pole body in yeast (Kaganovich et al, 2008). In contrast, in 2009, Wang and colleagues used co-localization with the MTOC protein Spc72 to establish that Htt inclusions in yeast are localized to the MTOC, and they further concluded that aggregated Htt is brought to the inclusion by microtubule-based transport, analogous to the mammalian aggresome (Wang et al, 2009).

To clarify the relationship between the Htt IB and the MTOC, we tagged the spindle-pole body proteins Spc72 and, separately, Spc42

with mCherry. Spc42-GFP containing MTOCs have been examined by electron microscopy and shown to be normal (Muller et al, 2005). We found no evidence that either mHtt(72Q)-GFP or mHtt(103Q)-GFP inclusion bodies localize to the MTOC, using either Spc42-mCherry or Spc72-mCherry as an MTOC marker (Fig S7A–C). These findings are consistent with the observed mobility of the IB (Fig 2).

As described above, previous work has suggested that mHtt is localized to IPODs, and that IPODs contain the autophagosomal protein Atg8. It has further been proposed that IPODs may be removed into the vacuole by macroautophagy (Kaganovich et al, 2008). Atg8 is considered a characteristic component of IPODs, and autophagy is frequently proposed as a mechanism of removal of aggregated protein in yeast and other organisms (Pankiv et al, 2007; Jeong et al, 2009; Tyedmers et al, 2010; Chen et al, 2011; Specht et al, 2011; Wong et al, 2012; Lu et al, 2014; Sontag et al, 2014; Miller et al, 2015). However, the co-localization of the IPOD with GFP-Atg8 was performed on cells that had been heat-shocked (Kaganovich et al, 2008). We co-expressed GFP-Atg8 and mHtt-mCherry in non–heat-shocked cells, to quantify the fraction of mHtt inclusions that contain Atg8. In a sample of 676 cells, 294 of which contained Htt inclusion bodies, we found no evidence of overlap between Atg8 and Htt aggregates (Fig S8). In addition, we introduced mHtt(72Q)-GFP into cells carrying a deletion of the gene encoding the vacuolar protease Pep4. Strains carrying deletions of *pep4* have been shown to accumulate GFP-tagged proteins in the vacuole (Welter et al, 2010; Delorme-Axford et al, 2015; Torggler et al, 2017). We examined *pep4Δ* cells expressing mHtt-GFP and compared them with BY4741 expressing mHtt-GFP; we saw no evidence of mHtt-GFP accumulation in the vacuole in *pep4Δ* cells (n = 125, 113 cells, respectively).

The absence of detectable co-localization with Atg8 and failure of mHtt-GFP to accumulate in the vacuole of *pep4Δ* cells is consistent with the direct observation of the endurance of inclusions in growing cells: in 360 h of imaging IBs in actively growing cells, we never saw an IB disappear from a cell.

## Discussion

We have investigated the structure, dynamics, and long-term fate of mHtt-GFP inclusions in yeast constitutively expressing mHtt(72Q)-GFP. We find that the large mHtt IB constitutes a non–membrane-bounded compartment with a set of characteristics unlike those of previously described unfolded protein inclusions, such as the aggresome or IPOD, categories to which the Htt IB has previously been assigned.

Earlier studies used Western blot analysis to show that yeast cells expressing mutant Htt-GFP contain aggregated mHtt-GFP, with the load of insoluble protein increasing with the length of the polyglutamine tract (Krobitsch & Lindquist, 2000; Muchowski et al, 2000). Visualization of mHtt-GFP in these strains shows that the prevalence of large cytosolic inclusions of mHtt-GFP is also proportional to polyglutamine tract length (Krobitsch & Lindquist, 2000). Although the assumption has been that the protein in large inclusions is highly insoluble, we find that the mHtt-GFP inside large inclusions is not fixed in place, as one might presume for a large body of aggregated protein. Our expectation was that new mHtt-GFP added to inclusions would accrete on the

outside of the IB, but instead, our FRAP and photoconversion studies show that newly incorporated protein appears throughout the entire volume of the inclusion.

In addition, Htt inclusions have smooth edges and a high circularity index and are mobile, all of which are characteristic of phase-separated liquid compartments. However, unlike in liquid droplets, the FRAP in these inclusions takes place over minutes rather than seconds. The timescale is comparable with the fluorescence loss observed as GFP-tagged proteins diffuse out of a gel (Kato et al, 2012). Given that we have observed neither division of IBs, nor coalescence in cells with multiple IBs (manuscript in preparation), we do not believe that mHtt IBs are fully liquid. These observations suggest that inclusions may have a gel-like consistency.

Cells expressing mHtt(72Q)-GFP typically contain a variable number of small mHtt particles. Analysis of the movement of small particles of aggregated mHtt shows that they diffuse randomly and are not actively transported. The larger inclusion bodies show a degree of active transport, but their movement is not visibly directional. Therefore, it is possible that the IB picks up aggregated protein through random collision as IBs and particles of aggregated protein diffuse through the cytosol (Fig 7).

The collision and coalescence model is consistent with the inverse correlation between small particle number and the presence and size of an IB, as collisions between IBs and small particles would be expected to increase with the surface area of the IB. The deletion mutants h*sp104Δ* and *rnq1Δ*, which lack small particles, also fail to form IBs. Last, 3D rendering of IBs shortly after photobleaching, as they begin to accumulate new fluorescent material, shows a nonuniform distribution of fluorescence suggestive of incorporation of small particles.

What is the function of the IB? The observation that unstructured proteins may be taken into a phase-separated compartment for renaturation or release has precedents. Others have shown that stress granules are not removed through autophagy, but are disassembled through the coordinated action of a complex that includes chaperones Hsp70 and HspB8, along with BAG3 (Ganassi et al, 2016). Mammalian cells also express a disaggregation complex, made up of Hsp110, Hsc70, and Hsp40. This chaperone combination has been shown to renature disordered and amyloid aggregates in vitro (Rampelt et al, 2012; Gao et al, 2015). Hsp110 is expressed at particularly high levels in the brain (Hylander et al, 2000). The

addition of Hsp104 to the mammalian disaggregase complex significantly increases its effectiveness in vitro (Shorter, 2011), and Hsp104 has been shown to reduce mHtt aggregation and lengthen life in transgenic mice that express mHtt(82Q) (Vacher et al, 2005).

Hsp104 is required for the formation of small particles of aggregated mHtt, and also Htt inclusions, suggesting that small particles are required for the formation of IBs. In vitro biochemical assays demonstrate that Hsp104 catalyzes the formation of amyloid fibers at low concentration but unwinds them at high concentration (Shorter & Lindquist, 2004).

Consistent with this idea, we have shown that mHtt is incorporated into a mobile Hsp104-containing inclusion. A pulse-chase experiment using photoconverted mHtt-GFP protein demonstrated that the inclusions undergo protein flux, with molecules both entering and leaving over time. We have directly measured for the first time the in vivo release of protein from an inclusion containing a disaggregase and protein aggregates. Because mHtt inclusions were not removed as a unit from the cell, our results support the idea of a dynamic cellular compartment involved in protein refolding.

As Hsp104 is concentrated significantly above cytoplasmic levels in the IB, we propose a model in which Hsp104 catalyzes the formation of mHtt fibers in the cytoplasm, and unwinds them in the inclusion. The decrease in cytosolic photoconverted mHtt-mEos2 demonstrates that the mHtt is ultimately degraded. However, it remains to be determined whether Hsp104 is definitively involved in the release of protein from the inclusion. Hsp104 has been implicated in the in vivo release of protein from aggregates in thermally stressed cells (Klaips et al, 2014; Kroschwald et al, 2018). Further studies will investigate whether Htt protein is degraded within the inclusion or outside it, perhaps by the ubiquitin–proteasome system, and the role of the Hsp104 in that process.

Unlike IPODs, mHtt inclusion bodies exchange material with the cytoplasm, but do not contain Atg8. As IBs have not been observed to disappear from untreated, actively growing cells, it is clear that they are not autophagocytosed into the vacuole under normal conditions. Instead, they constitute a novel, mobile, non–membrane-bound, phase-separated compartment from which mHtt protein is continuously removed. This is consistent with the idea that inclusion bodies are cytoprotective rather than toxic and suggests a new model for the function of inclusion

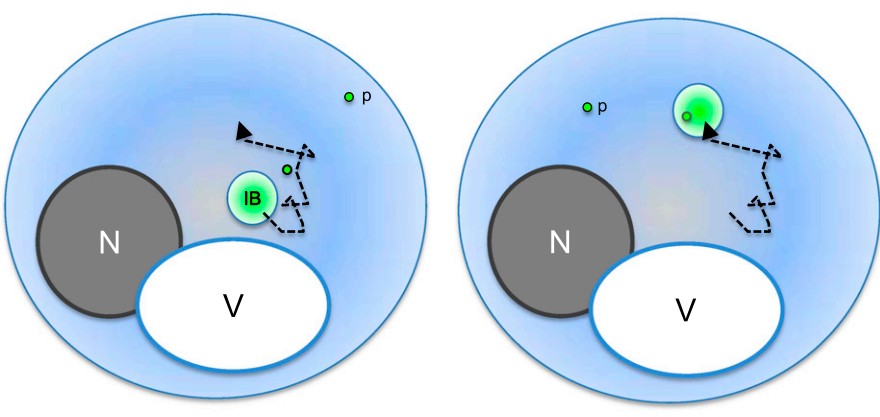

**Figure 7. Mutant Htt IBs may incorporate smaller particles of aggregated protein by collision.**
The simplest model of mHtt IB growth that is consistent with direct observations of IB and small particle movement is that IBs incorporate small aggregates of mHtt through collision as they diffuse in the cytosol. Mutant Htt IBs are constantly moving and, in contrast to IPODs, do not contain Atg8. Nucleus (N), vacuole (V), IB, two small mHtt particles (p), and IB path (dashed black line) are indicated. A small particle is shown being incorporated into the IB after the two collide.

formation, potentially relevant to diseases characterized by protein aggregation.

What is the mechanism underlying phase separation? The phenotype of $rnq1\Delta$ cells suggests that Rnq1 is an LCR protein whose normal function is to mediate the formation of a subset of Hsp104-dependent inclusions. The role of Rnq1 in forming gel-like, phase-separated compartments in vivo is consistent with the behavior of several LCR proteins in vitro: a number of proteins with regions of low-complexity sequence have been shown to spontaneously form hydrogels composed of amyloid-like fibers (Kato et al, 2012; Murakami et al, 2015; O'Rourke et al, 2015). The ability of overexpressed Rnq1-GFP to aggregate has been shown to depend on Hsp104 and on wild-type Rnq1 itself, and Rnq1-GFP aggregation is modified by the absence of any one of >100 other proteins (Sondheimer & Lindquist, 2000; Manogaran et al, 2010). In a recent study, Peskett and colleagues used a strain that had been selected for the inability of overexpressed Rnq1-GFP to aggregate (Peskett et al, 2018). They observed a concomitant change in the physical properties of aggregated mHtt-GFP. We did not observe liquid assemblies of mHtt-GFP with such physical properties in the parental strain, nor in cells lacking Rnq1 (Fig S5). These observations are consistent with a central role for Rnq1 in determining the structural properties of normal mHtt-GFP inclusion bodies.

A number of important questions are raised by our observations. Mutant Htt(72Q)-GFP can form distributed, peripheral aggregates, such as those seen in cells with CLIs, but rarely does. Occasionally, a cell will contain multiple ovoid IBs. Given that ovoid IBs are not localized to a single location in the cell, but diffuse around the cytosol, and that smaller particles of aggregated protein are moving randomly, it is not clear how a single, ovoid IB is maintained. Why don't smaller aggregates fuse randomly to make multiple inclusion bodies? One possibility is that a single compartment is energetically favored and may form through a process such as Ostwald ripening, a process that favors growth of larger compartments and that does not require the coalescence of macroscopic bodies. Another possibility is that cells have a mechanism for initiating only one IB.

# Materials and Methods

### Strains, plasmids, and culture conditions

Yeast were cultivated following standard procedures (Sherman, 2002). Cultures were grown in the appropriate glucose-based selective media at 30°C with shaking, except where specified. Transformation was performed using the lithium-acetate method (Gietz & Schiestl, 2007) and transformants were selected on the appropriate selective medium.

Plasmids and primers used in this study are listed in Tables S1 and S2, respectively. Mutant and native huntingtin expression constructs pEB4, pEB9, and pEB11 were constructed as follows: Htt exon 1 was amplified from the appropriate Addgene plasmid and subcloned into p415GPD (Mumberg et al, 1995). All plasmid DNA sequences derived from a PCR amplicon were sequenced and found to be free of mutations that change coding or known regulatory sequences. Color variants of pEB4 (mHtt(72Q)-GFP) were

made by excising $gfp$ and replacing it with $mCherry$ or $mEos2$ (Shaner et al, 2004; McKinney et al, 2009).

Strains used in this study are listed in Table S3. BY4741 $MAT\alpha$ $his3\Delta1$ $leu2\Delta0$ $met15\Delta0$ $ura3\Delta0$ was the parental strain for all other strains used. To make integrated genomic $mCherry$ fusions, primers were designed as described by Young and colleagues (Young et al, 2012). Kapa Hifi (KapaBiosystems) DNA polymerase was used to fuse the C terminus of selected genes to $mCherry$, using plasmid pCY3090-02 as a template. Yeast strains were transformed with the PCR product directly. Colonies were screened using a combination of hygromycin sensitivity and mCherry expression.

Atg8 was N-terminally tagged with GFP using the plasmid pOM42 to insert the GFP-coding sequence into the 5′ end of the $ATG8$ locus and plasmid pSH62 to express Cre recombinase and excise the selectable marker (Gueldener et al, 2002; Gauss et al, 2005). Colonies were initially screened visually for GFP expression, and the full $GFP$-$ATG8$–coding sequence of GFP-positive clones was confirmed by PCR followed by sequencing.

Deletions of $hsp104$ and $rnq1$ were accomplished by amplifying the $kanmx4$ cassette from the deletion library strain, obtained from Open Biosystems. Colonies were selected using G418, and PCR amplification was used to confirm that $kanmx4$ had replaced the gene loci. The $pep4$ deletion strain from Open Biosystems was cultured on G418 plates and used for transformations with pEB4 and pEB9.

### Epifluorescence image acquisition and data analysis

Yeast cultures were grown in selective minimal media to mid-log phase ($OD_{600}$ 0.1–0.3) or postdiauxic ($OD_{600} \geq 1.5$). Actively growing cells were grown for at least 10 doubling times, to mid-log ($OD_{600}$ = 0.1–0.3). On average, 69% of the actively growing cells were budding. Postdiauxic cells exiting growth were grown in selective synthetic minimal medium to an $OD_{600}$ >1.5. 36% of the cells were budding on average. For quantification of inclusion size, live cells were mounted under a #1.5 coverslip on a slide and imaged using an Axioskop 2 microscope (Carl Zeiss Inc.) with a 100×/1.40 Plan Apochromat oil immersion objective, an ORCA-ER charge-coupled device camera (Hamamatsu) and NIS Elements software (Nikon Instruments). Z-stacks were acquired with 200-nm spacing. To avoid bias in selection of cells, fields of view were selected in transmitted-light mode. Because we observed significant photoconversion of GFP to the red form (Brejc et al, 1997; Thor, 2011) after excitation with blue light on this microscope, the mCherry image was always acquired before the GFP image. For each genotype, two to four independently derived transformants were imaged and quantified.

Quantification was performed on unprocessed images using the Fiji distribution of ImageJ (Schindelin et al, 2012; Schneider et al, 2012). To measure inclusion size, the z-slice that showed maximum intensity in the inclusion was selected. The mean cytoplasmic intensity of the cell was measured using a region of interest drawn in the cytoplasm, but excluding the inclusion and vacuole. The image was thresholded to 1.2 times the mean cytoplasmic intensity, and the "Analyze Particles" function was then used to find the cross-sectional area of the IB. Fluorescent bodies visible in more than one focal plane and below the intensity threshold for IBs, or above the intensity threshold for IBs but with an area less than

0.01 $\mu m^2$, were counted as small particles. Circularity measurements were made on thresholded IBs using the ImageJ function *Shape Descriptors*.

We calculated the probability of apparent chance overlap between the IB and autophagosome as follows: at the central focal plane of a cell, we measured the cross-sectional area of the cell and, separately, the vacuole and subtracted the area of the vacuole from the total area of the cell for the approximate cytoplasmic area available to IBs and autophagosomes (n = 18). We measured the diameter of 80 mHtt IBs, which are variable in size; the mean diameter was 0.73 $\mu m$, and the median diameter was 0.48 $\mu m$. We measured the apparent diameter of Atg8-labeled autophagosomes, which are much less variable in size (mean and median diameters are 0.26 $\mu m$, n = 26). For any particular IB, the autophagosome will appear to be touching or overlapping if, by chance, it is within a circle whose diameter is equal to the diameter of the IB plus twice the diameter of the autophagosome. The area of the circle was calculated as A = $\pi r^2$ using the mean radius of autophagosomes and the median or mean radius of Htt(72Q) IBs and was found to be 5% or 8%, respectively, of the total available area of the average cell.

### Particle tracking and analysis

For imaging aggregate motility in mHtt-GFP cells, mid-log cells were imaged using a 100×/1.49 Apo-TIRF objective lens on a TiE Eclipse microscope (Nikon) equipped with a CSU-X1 spinning-disk unit (Yokogawa Electric), and a Zyla sCMOS camera (Andor). The cells were mounted on a 2% agar pad made with selective media and covered with a #1.5 coverslip. Time-lapse images were acquired at 30–32 fps. To image aggregates in mEos-expressing cells, mEos was photoconverted using a 5-s exposure to the 405-nm laser before imaging. Time-lapse images were acquired at 9–10 fps.

Time-lapse images were denoised with the Advanced Denoising function of NIS Elements Advanced Research software v.4.6, using the "original" algorithm and a denoising power of 10. Denoised movies were cropped so that they contained a single particle (aggregate or IB) visible for a minimum of ~1 s. Particles were tracked using the SpotTracker plugin (Sage et al, 2005) for ImageJ. Briefly, the image of the particle was further enhanced using the SpotTracker Spot Enhancing Filter 2D; with a minimum pixel size set to two for small particles and four for IBs. The SpotTracker 2D tracking algorithm was applied to the enhanced movie, using the same parameters for all particles. The parameters used to identify the mHtt particles were as follows: a maximum particle displacement of 10 pixels, weights on intensity factor set to 80%, intensity variation to 20%, movement constraint to 40%, and center constraint to 0%. Successful particle path identification was confirmed by comparing the original time-lapse data to the particle path file saved by SpotTracker.

To calculate mean squared displacement, the distance traveled in each possible time interval was calculated from the particle coordinates determined by SpotTracker. Average displacement was calculated for intervals of 1 frame (~30 ms/100 ms) to 16 frames (~480 ms/1,600 ms). Average time intervals were determined from the image timestamps.

The diffusion constant was calculated as follows: For a particle diffusing in 2D, where d is the distance traveled in time t and D is the diffusion constant,

$$d(t)^2 = 4Dt;$$
$$\text{therefore, } D = d(t)^2/4t.$$

To derive $\alpha$, we plotted $\log_{10}(d^2)$ versus $\log_{10}t$ and fitted a straight line in Excel (Microsoft); the value of $R^2$ was >0.98 for all fitted lines.

### FRAP imaging and analysis

Live cells were mounted on a 2% agar pad made with selective minimal media, covered with a #1.5 coverslip, and imaged using a 100×/1.45 Plan-Apo $\lambda$ objective lens on a TiE Eclipse microscope with A1 confocal scanner and GaAsP detectors (Nikon). The IB was photobleached using the 488-nm laser and recovery of fluorescence was followed by collecting z-stacks using a stage-mounted piezoelectric focus drive (Mad City Labs) over a depth of 3.6 $\mu m$ at 100-nm intervals, every 15 s for 10 min or every 2 min for 20 min. Focus was maintained using the Perfect Focus System (Nikon). Pinhole diameter was set to 0.6 Airy units; the predicted optical section for our imaging parameters is 0.30 $\mu m$ and the XY resolution is 0.18 $\mu m$, using Nikon Elements Confocal software. ImageJ was used to quantify fluorescence recovery. Intensity measurements were performed on a maximum-intensity projection. Cytoplasmic intensity was corrected for background and was measured in an area of the cytoplasm that was distant from the IB and did not include the vacuole. Integrated IB intensity was calculated as the product of area and mean intensity. IB perimeter was defined by thresholding to 1.4 times the mean cytoplasmic intensity. Mean IB intensity was corrected for photobleaching as described below. 3D rendering of the IB was performed in NIS Elements using the Maximum Intensity algorithm and a Z-zoom of 75%.

### Photoconversion imaging and analysis

Live cells were mounted on a 2% agar pad made with selective minimal media, covered with a #1.5 coverslip, and imaged using a 100×/1.45 Plan-Apo $\lambda$ objective lens on a TiE Eclipse microscope with A1 confocal scanner and GaAsP detectors (Nikon). mHtt-mEos2 was photoconverted from green to red using the 405-nm laser and cells were imaged sequentially line by line in green and red channels. Z-stacks were collected over a depth of 6.0 $\mu m$ at 300-nm intervals, every 10 min for 4.5 h.

To determine loss of signal due to photobleaching, fresh fields of cells were photoconverted and imaged for 10 time-lapse cycles with the same imaging parameters, except that the time-lapse interval was set to 0, that is, the fields were imaged as quickly as possible. Using ImageJ, maximum-intensity projections were made. For each series, mean red and green cytoplasmic intensity over time was measured on multiple cytoplasmic regions. These regions were chosen such that they did not undergo intensity changes due to movement of the vacuole and IBs. The average cytoplasmic intensities, normalized to the initial image, were fitted with exponential decay curves using Microsoft Excel, and the decay curve was used to predict bleaching behavior for time-lapse experiments. To

correct for bleaching, the fraction of intensity remaining after imaging was calculated from the bleaching decay curve for each round of imaging, and the corrected intensity was then calculated by dividing the measured image fluorescence intensity by the fraction of intensity remaining after bleaching.

Intensity measurements of the experimental images were made on a maximum-intensity projection using ImageJ. IB perimeter was determined by thresholding the green and red images to 1.5 times the mean cytoplasmic intensity, respectively, and then taking the union of the two masks. Red and green mean IB intensities were measured within that area. For each channel and time point, background cytoplasmic intensity was subtracted and mean intensity was corrected for photobleaching as described above. Integrated IB intensity was calculated as the product of the area and adjusted mean intensity.

### Immunogold labeling and transmitted electron microscopy

Fixation and immunogold labeling were performed as described by Mullholand and colleagues (Mulholland & Botstein, 2002). In brief, cells were grown overnight in synthetic complete (SC) or SC-Leu as appropriate and were fixed for an hour with 4% paraformaldehyde and 0.4% glutaraldehyde buffered with 0.08 M potassium phosphate with 0.5 M sorbitol added for osmotic support. The fixed cells were treated briefly with 1% sodium metaperiodate followed by 50 mM ammonium chloride. Immediately following fixation and periodate treatment, the cells were dehydrated in increasing concentrations of ethanol at 4°C and then embedded in LR White resin.

Fixed and embedded cells were cut to 50–90-nm thick-sections using an Ultracut UCT (Leica Biosystems) and collected on 200-mesh nickel grids. Sections were blocked with ovalbumin in TBST, then immunolabeled first with an anti-GFP antibody (Abcam 6556), followed by protein A conjugated to 10-nm gold particles (Electron Microscopy Sciences 25284). The sections were briefly fixed to preserve the position of the antibodies and stained with 2% uranyl acetate and, briefly, with 0.4% lead citrate.

Electron microscopy was performed using a JEM2100 transmission electron microscope (JEOL) with a point-to-point resolution of 0.23 nm and a line resolution of 0.14 nm. Images were acquired using an Ultrascan 1000XP and Digital Micrograph, Gatan Microscopy Suite (Gatan).

# Supplementary Information

# Acknowledgements

The authors would like to thank Istvan Boldogh, Ryo Higuchi-Sanabria, Enrique Garcia, E Laura Munteanu, Adrianne Mediavilla, Chevanie Bailey, and Jasodra Ramlall for their valuable advice and assistance with strain construction. We would also like to thank Art Palmer, Liza Pon, Chris Rongo, and Cecilia Östlund for their insightful comments on the manuscript. This work was supported by grants from the National Institutes of Health (NIH) to L Emtage (SC2GM116697), from the Professional Staff Congress of the City University of New York to L Emtage (Award # 68262-00 46), and from Pace University to C Burudpakdee and L Emtage. Images were collected and image processing and analysis for this work were performed in the Confocal and Specialized Microscopy Shared Resource of the Herbert Irving Comprehensive Cancer Center at Columbia University, supported by NIH grant #P30 CA013696 (National Cancer Institute). The confocal microscope used for photobleaching studies was purchased with NIH grant #S10 RR025686. Transmission electron microscopy was performed at the City College (City University of New York) Core Facility using the JEOL 2100 with assistance from Jorge Morales.

## Author Contributions

F Aktar: investigation.
C Burudpakdee: conceptualization and resources.
M Polanco: investigation.
S Pei: investigation.
TC Swayne: software, formal analysis, and writing—review and editing.
PN Lipke: conceptualization and writing—review and editing.
L Emtage: conceptualization, data curation, formal analysis, supervision, funding acquisition, validation, investigation, visualization, methodology, project administration, and writing—original draft, review, and editing.

## Conflict of Interest Statement

The authors declare that they have no conflict of interest.

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
