## [Reviewer comments · Life Science Alliance]

Life Science Alliance

The huntingtin inclusion is a dynamic phase-separated compartment

Fahmida Aktar, Chakkaong Burudpakdee, Mercedes Polanco, Sen Pei, Theresa Swayne, Peter Lipke, and Lesley Emtage

DOI: <https://doi.org/10.26508/lsa.201900489>

Corresponding author(s): Lesley Emtage, City University of New York, York College

Review Timeline:	Submission Date:	2019-07-17
	Editorial Decision:	2019-07-18
	Revision Received:	2019-08-12
	Editorial Decision:	2019-08-20
	Revision Received:	2019-08-22
	Accepted:	2019-09-02

Scientific Editor: Andrea Leibfried

Transaction Report:

Please note that the manuscript was previously reviewed at another journal and the reports were taken into account in the decision-making process at Life Science Alliance.

Reviewer #1 Review

Comments to the Authors (Required):

In this paper, Aktar et al. present data to support their claim that huntingtin (Htt) inclusions (comprised of Htt with a polyglutamine expansion) are dynamic phase-separated compartments. They further claim that Hsp104 is continuously liberating Htt from these structures, and that they are the first to show protein disaggregation by Hsp104 in yeast cells. Unfortunately, these claims are undercut by prior studies that have already established that Htt with a polyQ undergoes liquid-liquid phase separation and a pathological transition to solid states in vitro and in vivo (Pesket et al., 2018). This work is not cited. Moreover, the paper is further undermined by several studies showing that Hsp104 can indeed drive protein disaggregation and dispersal of phases in vivo (Cherkasov et al., 2013; Klaips et al., 2014; Kroschwald et al., 2018; Kumar et al., 2016; Parsell et al., 1994). None of these works are cited, indicating poor scholarship. However, one paper that shows Hsp104 can drive protein disaggregation in vivo is cited (Kroschwald et al., 2015), but not credited for showing Hsp104-driven protein disaggregation in vivo. Given these issues of novelty and scholarship, I suggest that this work is not suitable for this journal and better suited to a more specialized journal.

Cherkasov, V., S. Hofmann, S. Druffel-Augustin, A. Mogk, J. Tyedmers, G. Stoecklin, and B. Bukau. 2013. Coordination of translational control and protein homeostasis during severe heat stress. *Curr Biol.* 23:2452-2462.

Klaips, C.L., M.L. Hochstrasser, C.R. Langlois, and T.R. Serio. 2014. Spatial quality control bypasses cell-based limitations on proteostasis to promote prion curing. *Elife.* 3.

Kroschwald, S., S. Maharana, D. Mateju, L. Malinowska, E. Nuske, I. Poser, D. Richter, and S. Alberti. 2015. Promiscuous interactions and protein disaggregases determine the material state of stress-inducible RNP granules. *Elife.* 4:e06807.

Kroschwald, S., M.C. Munder, S. Maharana, T.M. Franzmann, D. Richter, M. Ruer, A.A. Hyman, and S. Alberti. 2018. Different Material States of Pub1 Condensates Define Distinct Modes of Stress Adaptation and Recovery. *Cell Rep.* 23:3327-3339.

Kumar, R., P.P. Nawroth, and J. Tyedmers. 2016. Prion Aggregates Are Recruited to the Insoluble Protein Deposit (IPOD) via Myosin 2-Based Vesicular Transport. *PLoS Genet.* 12:e1006324.

Parsell, D.A., A.S. Kowal, M.A. Singer, and S. Lindquist. 1994. Protein disaggregation mediated by heat-shock protein Hsp104. *Nature.* 372:475-478.

Pesket, T.R., F. Rau, J. O'Driscoll, R. Patani, A.R. Lowe, and H.R. Saibil. 2018. A Liquid to Solid Phase Transition Underlying Pathological Huntingtin Exon1 Aggregation. *Mol Cell.* 70:588-601 e586.

Reviewer #2 Review

Comments to the Authors (Required):

In this study, the authors employ a well-characterized yeast model of Huntington's disease to study the dynamics of mutant Huntington (mHtt) aggregate formation. Surprisingly, they found that these aggregates are dynamic with mHtt constantly being removed from the aggregates using live cell imaging techniques. This process is proposed to involve the disaggregase protein Hsp104. mHtt aggregates are different from IPOD and aggresomes. They conclude that mHtt inclusion bodies grow as a result of collision and coalescence with diffusing aggregate particles.

While I think there are some interesting observations in the manuscript, I have several concerns that at this moment prevent its publication in this journal.

Major concerns:

1-One of the major concerns is about the major finding of the paper that mHtt is mobile within the aggregates. In a recent paper in *Molecular Cell*, Peskett et al., showed that aggregates are rather immobile in bright aggregates. The authors should discuss the discrepancies between the two studies. The Peskett paper was not even cited.

2-The authors should comment on the constructs employed. In yeast, polyQ toxicity and structure of the aggregates depend on the presence of the proline-rich domain. What would happen to the mobility if you use the toxic version? That would be more relevant to the disease and add important information to the model proposed.

3-In Figure 4, data in B are not that convincing. Maybe it would be better if presented as % of maximum over time. Also how's the protein turnover? Repeating the experiments in cells lacking either autophagy, proteasome or vacuole activity would be much more convincing and would shine light onto the mechanisms. Also I would add the green channel data (Figure S3) to the main figure.

4-In Figure 5, it is known that HSP104 deletion impedes aggregate formation by preventing prion propagation. Is the formation of the small particle dependent on Rnq1?

5-Bottom line is that the study shows new properties of the mHttex1 aggregates in yeast but not much mechanistic details. They also failed to reproduce previous studies, which this reviewer thinks it's perfectly ok, but they should provide context and interpretation as to why the outcome is different (localization to IPOD and atg8).

6-Minor concern:

-Please add page and line numbers on the manuscript.

-Since fluorescent proteins have been shown to affect mHtt aggregation and toxicity in yeast, is the mEos2 equivalent to GFP in terms of the ability to form aggregates? Are the proteins as mobile?

July 18, 2019

Re: Life Science Alliance manuscript #LSA-2019-00489-T

Dr. Lesley Emtage
City University of New York, York College
Biology
94-20 Guy R. Brewer Blvd.
New York, New York 11451

Dear Dr. Emtage,

Thank you for transferring your manuscript entitled "The huntingtin inclusion is a dynamic phase-separated compartment" to Life Science Alliance. The manuscript was assessed by expert reviewers at another journal before and the editors provided those reports to us with your permission.

The reviewers thought that your work is not properly placed into the context of the existing literature and that some more insight would be required to provide knowledge on mHtt dynamics of value to others.

Based on this input, we would like to invite you to submit a revised version to us. We'd expect a point-by-point response to all concerns raised, accordingly proper placing of the work into the existing literature as well as adding some more insight by following the suggestions made by ref#2 (points 2-4). We are aiming at engaging with the same reviewer #2 for re-review to enable an efficient process. We will of course explain the transfer situation to the reviewer upon re-review.

Thank you for this interesting contribution to Life Science Alliance. We are looking forward to receiving your revised manuscript.

Sincerely,

Andrea Leibfried, PhD
Executive Editor
Life Science Alliance
Meyershofstr. 1
69117 Heidelberg, Germany
t +49 6221 8891 502
e a.leibfried@life-science-alliance.org
www.life-science-alliance.org

B. MANUSCRIPT ORGANIZATION AND FORMATTING:

1. Focus and clarity: We have rewritten the abstract to more clearly emphasize our findings regarding the physical nature of the inclusion, the mobility of the inclusion and small particles, and the proposed model.
2. Role of Hsp104: Hsp104 has been shown to undergo a switch between catalyzing the formation of and the disaggregation of amyloid fibers, depending on the relative concentrations of Hsp104 and its substrate (Shorter and Lindquist, 2004). The requirement for Hsp104 in formation of mHtt inclusion bodies has already been shown (Krobitsch and Lindquist, 2000); we show here that it is also required for the formation of small particles, and that it is found in mHtt inclusions and small particles. Because we directly observe release of mHtt from inclusions, we believe it is reasonable to suggest that when Hsp104 is concentrated in the inclusion, it may switch activities to release material from the inclusion. However, because there are no inclusions in *hsp104Δ* cells, we cannot speak directly to a role for Hsp104 in release of protein from the inclusion. Therefore, in our revised manuscript, we have de-emphasized the proposed role of Hsp104 by removing it from the abstract, although it is still discussed.

To provide further background and clarify the diverse roles of Hsp104, we have added a summary of previous work concerning the activity of Hsp104 in other types of aggregative compartments to the Discussion. The role of Hsp104 appears to be different in heat-induced responses to stress versus mHtt inclusions, as stress granules have been reported to form in *hsp104Δ* cells (Kroschwald, 2015), whereas mHtt inclusions do not form in these cells (Krobitsch and Lindquist, 2000; Meriin et al., 2002).

3. Reviewer 2, point 1:

One of the major concerns is about the major finding of the paper that mHtt is mobile within the aggregates. In a recent paper in Molecular Cell, Peskett et al., showed that aggregates are rather immobile in bright aggregates. The authors should discuss the discrepancies between the two studies.

We have added a discussion of the Peskett paper to the Discussion, and we also address issues raised by this paper in the Results sections on FRAP and characterization of *rnq1Δ* cells.

Although Peskett and colleagues also investigated mHtt dynamics using time-lapse and photobleaching techniques, their experimental system and design were quite different from ours. They expressed mHtt(96Q)-GFP from a 2-micron high-copy plasmid (40-60 copies per cell) under the control of the GAL promoter, in cells grown in galactose. In contrast, we expressed mHtt(72Q)-GFP constitutively using a low-copy CEN plasmid (1-5 copies, Karim et al. (2013), doi: 10.1111/1567-1364.12016). Thus, the size of the glutamine expansion, the carbon source, and the level of protein expression were different from those used in our studies. Possibly as a result of this, the frequency and

morphology of the inclusions they observe appear also to be different. For example, in Peskett et al., Figure 2, three out of four cells shown have very large asymmetric inclusions. Such a situation would be quite atypical in our strains and may be the result of expressing mHtt(96Q) from a high-copy plasmid under the control of the GAL promoter. An alternative possibility is that the images represent cells in late-log/postdiauxic cultures, which exhibit increased size and frequency of inclusions (see our revised Figure S1). We have also found that growth in galactose, in addition to slowing culture growth, can affect inclusion formation and morphology, and thus we have used other carbon sources in our work.

For their studies of mHtt(96Q) dynamics, Peskett and colleagues report values for fluorescence recovery in bright inclusions for only 20 seconds post-bleach, imaging approximately 180 times (9 fps) while monitoring recovery. In contrast, we imaged every 15 seconds (reducing additional bleaching due to imaging). In our system, we begin to see recovery of fluorescence into ovoid inclusions at about 30 seconds, and see recovery throughout the majority of the inclusion volume by 10-20 minutes post-bleach. This could be because of the less invasive imaging regime, or because the inclusions formed in the two experimental systems are structurally different.

The distinction between different types of inclusions is a difficult one to make, in part due to the lack of molecular markers. Based on the frequency and shape of the inclusions shown in Peskett, Figure 2, we may be doing FRAP on different types of intense inclusions. But, equally, it is not possible to know whether their RNQ+ mHtt inclusions would eventually recover fluorescence at longer timepoints, as they only report recovery to 20 seconds post-bleach.

4. Reviewer 2 (point 2) asked for a comment on the constructs employed. We have added a description of our constructs to the Results section, in addition to the description in the Methods.
5. Reviewer 2, point 2:

In yeast, polyQ toxicity and structure of the aggregates depend on the presence of the proline-rich domain. What would happen to the mobility if you use the toxic version? That would be more relevant to the disease and add important information to the model proposed.

We have measured growth in cells expressing a mHtt(72Q) Δ Pro-GFP variant and confirmed that it (a) shows reduced growth, and (b) fails to form the typical ovoid inclusion bodies seen in cells expressing mHtt(72Q)-GFP. Rather, it forms many smaller 'distributed' inclusions, scattered throughout the cytoplasm, with some cells also having large, asymmetric inclusions. These observations are consistent with the original characterization of the Δ Pro construct (Dehay and Bertolotti, 2006).

Dehay and Bertolotti report large inclusions of Δ Pro in *hsp104* Δ cells; however, we have been unable to reproduce these findings. Our mHtt(72Q) Δ Pro-GFP construct forms

distributed inclusions in the BY4741 background, but forms no visible inclusions at all in *hsp104Δ* cells. (full sequence of construct contains no mutations; *HSP104* deletion confirmed by PCR; unpublished observations, L. Emtage).

Three points mitigate against basing a model on the mHtt(72Q) Δ Pro-GFP variant. First, we can find no described cases in which a human carries a mutant huntingtin allele lacking the proline-rich region, which significantly weakens the case for disease relevance (for overview, see <https://www.omim.org/entry/613004>). Second, huntingtin is ubiquitously expressed in humans throughout life, and most human tissues can cope with relatively high levels of mutant Htt. We therefore view our studies as an investigation into the *survival* mechanism of cells unusually burdened with unstable protein. Third, our study focuses on the nature and growth mechanism of normal, ovoid inclusion bodies. We can measure the movement of the distributed mHtt(72Q) Δ Pro-GFP inclusions, but because the normal, ovoid inclusion bodies fail to form in these cells, this information cannot shed light on the nature and growth mechanism of typical inclusion bodies formed by mHtt.

6. Reviewer 2 (point 3) suggested presenting Fig. 4B as % of maximum fluorescence over time. We have made the suggested change.

7. Reviewer 2 (point 3) writes:

Also how's the protein turn over? Repeating the experiments in cells lacking either autophagy, proteasome or vacuole activity would be much more convincing and would shine light onto the mechanisms. Also I would add the green channel data (Figure S3) to the main figure.

The eventual fate of included and cytoplasmic mHtt-GFP is, naturally, very interesting to us. We believe that a thorough and appropriately quantitated investigation is warranted, but would be outside the scope of the current work. Nevertheless, we do have some data that addresses this question.

We have also expressed both native [Q25] and mutant [Q72] Htt-GFP in a *pep4* deletion strain; we see no evidence that GFP accumulates in the vacuole in the cells expressing mHtt-GFP and we have incorporated these data into the revised manuscript. This is in addition to data and discussion already present in the MS showing that no inclusion was observed entering a vacuole in 120 inclusions tracked for an average of 3 hours each.

A manipulation that produces an overall block to the proteasome will disrupt the regulation of many hundreds of regulatory proteins and cannot easily be interpreted, as it leads to pleiotropic effects. Similarly, an overall block to autophagy would also alter many cellular processes. We expect that changes to the level of mHtt would be altered either directly or indirectly as formation of the mHtt inclusion depends on cellular aggregation and disaggregation machinery. Substantial further experimentation will be required to disentangle the possibilities.

We share the reviewer's interest in this subject, but feel that the question deserves a detailed genetic and biochemical analysis of the molecular mechanism underlying mHtt turnover, which will be more appropriately dealt with in a separate set of studies.

Additionally, the photoconversion experiment is a pulse-chase study of red Htt(72Q)-mEos2. It is natural to inquire about the levels of green protein, but since (1) it is not relevant to the pulse-chase study, and (2) cannot be accurately quantified due to ongoing synthesis of green Htt(72Q)-mEos2, we have included it in the supplemental data.

8. Reviewer 2 (point 4) asked whether Rnq1 is required for the formation of small particles (as shown for Hsp104 in Fig. 5). Yes, the formation of small particles of mHtt-GFP also depends on Rnq1. We have incorporated those data into the manuscript (Figure S5), and added a discussion of Rnq1.

9. Reviewer 2 writes:

Since fluorescent proteins have been shown to affect mHtt aggregation and toxicity in yeast, is the mEos2 equivalent to GFP in terms of the ability to form aggregates? Are the proteins as mobile?

These data have been added to the manuscript. Aggregates in mHtt-mEos2-expressing cells appear similar to those in mHtt-GFP-expressing cells. We have also measured the mobility of the mHtt-mEos2 aggregates: like mHtt-GFP aggregates, small particles of mHtt-mEos2 move randomly, the IBs show some degree of active transport.

August 20, 2019

RE: Life Science Alliance Manuscript #LSA-2019-00489-TR

Dr. Lesley Emtage
City University of New York, York College
Biology
94-20 Guy R. Brewer Blvd.
New York, New York 11451

Dear Dr. Emtage,

Thank you for submitting your revised manuscript entitled "The huntingtin inclusion is a dynamic phase-separated compartment". As outlined to you before, we asked one of the original reviewers (previous reviewer #2) who evaluated your work at another journal to assess the revised version of your manuscript. As you will see below, the reviewer appreciates the changes introduced in revision and we would thus be happy to publish your paper in Life Science Alliance pending final revisions necessary to meet our formatting guidelines:

- please make sure to mention all error bars (eg., fig 1G, 5C/D, S4C)
- Figure 5: please indicate the ** mentioned in the legends in the figure
- please check the author order in our submission system to make sure that the correct order is shown
- please add a title/short legend for each suppl table
- Fig 4A: the scale bar appears twice, please fix

A. FINAL FILES:

-- High-resolution figure, supplementary figure and video files uploaded as individual files: See our detailed guidelines for preparing your production-ready images, <http://www.life-science->

alliance.org/authors

B. MANUSCRIPT ORGANIZATION AND FORMATTING:

Sincerely,

Andrea Leibfried, PhD
Executive Editor
Life Science Alliance
Meyershofstr. 1
69117 Heidelberg, Germany
t +49 6221 8891 502
e a.leibfried@life-science-alliance.org

Reviewer #1 (Comments to the Authors (Required)):

The authors have addressed my previous concerns and the manuscript is now acceptable for publication.

September 2, 2019

RE: Life Science Alliance Manuscript #LSA-2019-00489-TRR

Dr. Lesley Emtage
City University of New York, York College
Biology
94-20 Guy R. Brewer Blvd.
New York, New York 11451

Dear Dr Emtage,

Thank you for submitting your Research Article entitled "The huntingtin inclusion is a dynamic phase-separated compartment". It is a pleasure to let you know that your manuscript is now accepted for publication in Life Science Alliance. Congratulations on this interesting work.

DISTRIBUTION OF MATERIALS:

Again, congratulations on a very nice paper. I hope you found the review process to be constructive and are pleased with how the manuscript was handled editorially. We look forward to future exciting submissions from your lab.

Sincerely,

Daniel Klimmeck

Daniel Klimmeck, PhD
Scientific Editor
Life Science Alliance